# TGF-β in Cancer: Metabolic Driver of the Tolerogenic Crosstalk in the Tumor Microenvironment

**DOI:** 10.3390/cancers13030401

**Published:** 2021-01-22

**Authors:** Roberta Angioni, Ricardo Sánchez-Rodríguez, Antonella Viola, Barbara Molon

**Affiliations:** 1Department of Biomedical Sciences, University of Padova, 35131 Padova, Italy; roberta.angioni@unipd.it (R.A.); ricardo.sanchezrodriguez@unipd.it (R.S.-R.); antonella.viola@unipd.it (A.V.); 2Istituto di Ricerca Pediatrica IRP-Fondazione Città della Speranza, 35127 Padova, Italy

**Keywords:** immune-metabolism, tumor microenvironment, TGF-β

## Abstract

**Simple Summary:**

Metabolic reprogramming is an emerging hallmark in cancer. Beside the malignant compartment, the tumor microenvironment also undergoes to a metabolic skewing, contributing to the neoplastic progression and metastasizing process. Growing evidence pointed out a central role for Transforming Growth Factor Beta (TGF-β) as a driver of these metabolic changes in multiple cellular targets in cancer. This review deals with very recent discoveries on TGF-β-mediated metabolic reprogramming of stromal and immune cell population within the tumor microenvironment. In particular, we scrutinized current literature to highlight relevant metabolic checkpoints in the TGF-β cascade that sustain tolerogenic programs in tumors.

**Abstract:**

Overcoming tumor immunosuppression still represents one ambitious achievement for cancer immunotherapy. Of note, the cytokine TGF-β contributes to immune evasion in multiple cancer types, by feeding the establishment of a tolerogenic environment in the host. Indeed, it fosters the expansion and accumulation of immunosuppressive regulatory cell populations within the tumor microenvironment (TME), where it also activates resident stromal cells and enhances angiogenesis programs. More recently, TGF-β has also turned out as a key metabolic adjuster in tumors orchestrating metabolic pathways in the TME. In this review, we will scrutinize TGF-β-mediated immune and stromal cell crosstalk within the TME, with a primary focus on metabolic programs.

## 1. Introduction

### 1.1. The Tumor Metabolic Reprogramming

Corroborated evidence has highlighted the crucial contribution of the tumor microenvironment (TME) in cancer progression. The TME refers to the networks of nonmalignant cells and structures within the bulk tumor, mainly including fibroblasts, endothelial and immune cells. Tumor cells functionally shape the TME to establish a tolerogenic environment in the host. In turn, the TME sustains the structural and metabolic tumoral demands in a fixed pro-tumoral circuit. These events, based on multiple synergic processes, include the reprograming of both tumoral and TME metabolism [1]. The metabolic switch occurring in malignant cells during tumor progression has been largely documented. Indeed, under aerobic conditions, normal cells convert glucose to pyruvate in the cytosol (oxidative phosphorylation OXPHOS), thus obtaining energetic ATP molecules. In contrast, in anaerobic conditions, pyruvate is used to produce lactate in the cytosol through the glycolysis. Most malignant cells depend on glycolysis for the ATP production, even under aerobic conditions (aerobic glycolysis) [2]. This tumoral propensity to metabolize glucose anaerobically rather than aerobically is called Warburg effect. In 1920, Otto Warburg described for the first time that cancer cells have high rates of glucose uptake and lactate release even in the presence of oxygen [3]. Even though the first suggestions indicated an impairment in oxidative metabolism, consistent evidence later on showed that a tumoral metabolic reprogramming is responsible for an increase of glucose metabolism despite an intact respiratory chain activity [4]. These mitochondrial independent anabolic pathways, including pentose-phosphate based nucleotide and NADPH synthesis, and Krebs cycle intermediates contribute to maintain cancer progression [5]. Hypoxia-inducible factor (HIF) activation is a master regulator of the Warburg effect promoting the transcription of glycolytic genes [6], together with c-Myc [7] and SIX1 activity [8]. Beside the malignant compartment, the TME also undergoes to a metabolic reprogramming contributing to the neoplastic progression and metastasizing process. In addition, growing data pointed out a central role of TGF-β as driver of these metabolic changes in multiple cellular targets. Here, we will resume recent data on the TGF-β mediated TME metabolic reprogramming during tumoral development.

### 1.2. Transforming Growth Factor Beta Signaling in Tumors

TGF-β is a well know cytokine and growth factor related to tumor growth and fibrosis in different kind of cancers such as lung, pancreas, colon cancer and hepatocellular carcinoma [9]. Only recently, its effect on the tumor and TME metabolic reprograming has been revealed. TGF-β is a member of a protein family that includes activin, bone morphogenic proteins (BMP) and Nodal [10] and it is expressed in three different isoforms (TGF-β1, TGF-β2 and TGF-β3). TGF-β is synthetized as inactive form composed by a C-terminal domain and the latency-associated peptide (LAP) at the N-terminal. The circulating TGF-β binds through the LAP portion the latent TGF-β binding proteins (LTBPs) leading to the assembling of the large latent complex (LLC) [11]. The release of the active TGF-β from the LCC is dependent on the action of cell-surface transmembrane receptor proteins such as αvβ6 and αvβ8 integrins. Indeed, once bounded to the RDG sequence in the LAP, these proteins unfold the complex and induce the release of the active TGF-β [12]. Extracellular matrix (ECM) proteins involved in this process further include Thrombospondin 1 [13], Glycoprotein A repetitions predominant protein (GARP) [14], Leucine-rich repeat containing protein 33 (LRRC33) [15], Cathepsin D and metalloproteases (MMPs) such as MMP-9 and MMP14 [16]. Once released, active TGF-β directly associates to its heteromeric serine/threonine kinase receptor (TβRI-TβRII) [17], often assisted by accessory coreceptors as TβRIII or β-glycan [18]. Then, the TGF-β induced TβRII autophosphorylation, at the Ser213 and Ser409, determines its kinase activity on TβRI GS domain triggering two distinct downstream pathways, called canonical and non-canonical pathways [19]. In the SMAD-dependent canonical pathway, TβRI phosphorylates SMAD2 and SMAD3 to form an heteromeric complex with SMAD4. This complex accumulates into the nucleus where the expression of several genes is controlled by directly binding several DNA regulatory regions [20]. On the other side, the TGF-β non-canonical pathway activates GRB2, SOS or TAK1 leading to the MAPK kinases cascade that, in turn, regulates NF-κB, c-FOS, c-JUN c-MYC transcription factors [21]. Beside the transcriptional regulation, TGF-β-triggered circuits directly induce cytoskeleton remodeling by activation of small protein GTP-ases proteins such as RhoA, Rock or by Cofilin-1 signaling [22]. The duality of TGF-β signaling has been repeatedly documented in cancer as this cytokine acts as a potent tumor suppressor during early tumor development, but paradoxically, it become a metastatic promoter in later stages [23]. As extensively reviewed elsewhere [23], several studies have been developed to specifically manage this challenging target for cancer treatment [23].

Among many other documented effects, the canonical TGF-β/SMAD pathway has been linked to the epithelial-mesenchymal transition (EMT), that contribute to the metastatic spreading in not all, but many tumors. In pre-clinical breast cancer models the activation of SNAIL1-SMAD3/4 transcriptional complex regulated gene expression during EMT [24]. Further evidence showed that TGF-β–induced activation of mTOR and the PI3K–Akt pathway correlated with increased cell size and protein content in NMuMG mammary cells, supporting an important role for non-Smad TGF-β signaling pathway in cancer progression [25]. In this regard, a recent report indicated that a prolonged TGF-β exposure promoted stable EMT in mammary epithelial and carcinoma cells, by triggering the mTOR pathway; this stable EMT phenotype contributed to controlling cancer cell stemness drug resistance of breast cancer cells [26]. TGF-β cascades can differently act on multiple cell compartments within the bulk tumor, mainly establishing a pro-tumoral metabolic reprogramming of the TME components. In the TME, TGF-β can be produced by several cell types including tumor, stromal and immune cells mediating both autocrine and paracrine effects. Moreover, distant TGF-β sites of release can not be excluded in several cancers. Indeed, in most of the in vivo studies, technical impairments limited the precise definition of TGF-β release origin. In this manuscript, we will focus on the TGF-β impact on cancer-associated fibroblasts (CAFs), endothelial and immune cells, focusing on metabolic pathways (Figure 1).

## 2. TGF-β Induced Cancer-Associated Fibroblast (CAF) Metabolic Reprogramming

### 2.1. CAF General Function

CAFs are heterogenous cells of the TME. Their differences mainly depend on multiple origins, various tissue localizations and distinct activation stages [27]. Despite the precise CAF origin is still unsolved, collected evidence defined that CAFs might derived from: (i) activated resident fibroblasts, (ii) engrafted and differentiated mesenchymal stromal cells and (iii) accumulated mutations in epithelial or tumor cells [28]. This heterogeneity troubles the definition of specific CAF markers. Consequently, the precise CAF tissue localization is not fully described [29]. Furthermore, overwhelming data pointed out a sub-specialization of CAFs possibly reflecting distinct functional states [30]. Nonetheless, CAFs are usually identified by the expression of α-smooth muscle actin (α-SMA), fibroblast specific protein, fibroblast activation protein (FAP) and platelet-derived growth factor receptors α and β (PDGF-α and PDGF-β) [29,31]. These cells have displayed multiple pro-tumorigenic functions cooperating with other CAF-independent mechanisms [32]. In particular, CAFs exhibit either a matrix-producing phenotype or an immunomodulating secretome, with a biunivocal functional connection of these effects [30]. CAFs release ECM proteins (such as type I, type III, type IV and type V collagens, many different laminins and fibronectin) as well as extracellular enzymes (MMPs and lysyl oxidase) representing the main contributors of the ECM remodeling [33]. This CAF-sculpted tumor ECM composition and organization leads to increased matrix stiffness responsible for higher cancer cell adhesion to ECM, enhanced cancer cell proliferation and invasion, and elevated interstitial fluid pressure promoting pro-angiogenic signatures [34,35,36]. Concomitantly, CAFs shape the immune microenvironment toward a tolerant and immunosuppressive milieu contributing to cancer immune escape [37]. This process is dependent on the CAF production of both immunosuppressive cytokines and immune checkpoint ligands. Thus, CAFs block the cytotoxic CD8^+^ T cell recruitment, and prompt tumor infiltrating inflammatory cell polarization toward an anti-inflammatory phenotype [37]. Together, CAF-remodeled matrix and CAF-established immunosuppression within the tumor enhance cancer invasiveness, stemness and proliferative behavior, favoring EMT and sustaining tumor anchorage and metastaticity [27]. Besides these processes, CAFs further support the insurgence of a pro-tumoral microenvironment by engaging a metabolic interplay within the TME. Here, TGF-β has a central reprogramming role in both the (i) CAF generation and (ii) the delineation of a pro-tumoral TME.

### 2.2. How Does TGF-β Directly Alter CAF Functions?

A synergistic interplay between neoplastic cell transformation and the surrounding environment is required for detrimental cancer progression [38]. CAF activation is pivotal during the TME definition, and TGF-β family ligands play a crucial role in this process. The critical contribution of TGF-β to CAF generation is dependent on a metabolic reprogram, that mainly attends to an unbalance in the oxidative framework [30]. Indeed, the TGF-β canonical and non-canonical pathways increase reactive oxygen species (ROS) production in CAFs by the impairment of the respiratory transport chain, specifically acting on Complex IV [39] by the GS3K action [40]. Furthermore, Smad signaling contributes to increase the expression of NOX4 [41], thus sustaining ROS accumulation. Additionally, TGF-β-mediated transcriptional competition for several DNA-binding sites, as Glutamate-Cysteine Ligase (GSL) gene, determines glutathione depletion [42] and antioxidant NRF2 impairment [43], eventually contributing to the oxidative stress. ROS increase in fibroblasts modulates CAF markers as α-SMA expression [44] and it is linked to the loss of stromal caveolin-1 (Cav-1), due to an extreme induction of LC3B-II-mediated lysosomal degradation [45]. Beside the ROS-dependent mechanism, TGF-β activation increases the intracellular accumulation of α-ketoglutarate by downregulating the isocitrate dehydrogenase 1 (IDH1). This event further contributes to Cav-1 downregulation [46]. Notably, the loss of stromal Cav-1 within the tumor fuels cancer progression through various paracrine signaling mechanisms, including the initiation of CAF-mediated ECM remodeling [47]. Moreover, it has been reported that the attenuation of Cav-1 pushes fibroblasts toward an autophagic phenotype with elevate aerobic glycolysis. Indeed, the down expression of Cav-1 contributes to the mitochondrial disfunction thus creating a positive-loop between Cav-1 and ROS production [48]. Reduced expression of Cav-1 also triggers the down regulation and activity of prolyl hydroxylase domain-containing protein (PHD) [49], which is involved in the hydroxylation of HIF1α and IκB. The consequent HIF stabilization and NFκB activation leads to glycolytic gene transcription such as lactate dehydrogenase A (LDHA) and glyceraldehyde 3-phosphate dehydrogenase (GAPDH) [50]. All together, these steps determine the CAF secretion of high energy metabolites such as pyruvate and lactate compelling mitochondrial OXPHOS in cancer cells [51]. Along with Cav-1/glycolysis circuit, emerging data pointed out the TGF-β dependent autophagy-to-senescence transition (AST) of some TME component, especially CAFs [52]. In this scenario, autophagic fibroblasts enter a non-proliferative state termed “senescence” [53], characterized by enhanced glucose uptake and lactate efflux [54]. Here, fibroblasts acquire a senescence-associated secretory phenotype (SASP), depicted by a strong release of IL6, IL1, IL8, GROα and GROβ [55], which contributes to the generation of a tumor-promoting microenvironment [37]. Notably, a tumor cell senescence reinforcing-effect has been assessed on fibroblasts in response to release of TGF-β by recruited macrophages [56].

### 2.3. Does the TGF-β Dependent CAF Activity Shape the Tumor Metabolism?

Beside the TGF-β-dependent metabolic alteration of fibroblasts, responsible for CAF generation, TGF-β also triggers critical CAF metabolic functions in cancer [32]. Indeed, it has been proposed that, similarly to the Warburg effect in cancer cells, CAFs participate to the metabolic programming of the TME that further nourishes tumor-development [57]. In this scenario, cancer cells and CAFs become metabolically associated [58]. The metabolic crosstalk between stromal and cancer cells, where CAF-generated metabolites sustain the tumor metabolic requirement, takes the name of “reverse Warburg effect”. Here, cancer cells release hydrogen peroxide into the microenvironment, thus fostering oxidative stress in the adjacent CAFs. Concomitantly, CAFs undergo aerobic glycolysis and generate high level of energy-rich fuels (as lactate, pyruvate, ketone bodies and fatty acids), that feed cancer cells [57]. More insight, lactate is secreted by CAF through the monocarboxylate transporter 4 (MCT4), together with an H+ efflux. Therefore, a pH changes responsible for the EMT increase and MMP activity occurs in the TME [59]. In addition, monocarboxylate transporter 1 (MCT1) catalyzes the rapid transport of CAF-generated lactate across the plasma membranes of tumor cells [60]. Here, the concurrent uptake of CAF-released erythrose-4-phosphate leads to a sustained pyruvate production exploited as fuel for pentose phosphate pathway (PPP) [61]. Moreover, multiple ketone bodies accumulated from the CAF glycolytic metabolism, such as acetoacetate and β-hydroxybutyrate, provide a precious tumor energy source for Acetyl-CoA acetyltransferase and 3-hydroxy-3-methylglutaryl-CoA synthase activity during the ketone recycle [58,62]. The metabolic reprogramming in CAFs also induces a high release of glutamine used by tumor cells as nitrogen donor for amino acid synthesis [63]. Finally, CAF-related oxidative stress contributes to ROS accumulation within the tumor, thus supporting HIF and NFκB activation [64].

## 3. TGF-β Induced Endothelial Metabolic Reprogramming

### 3.1. Endothelial Cell (EC) General Function

Blood vessels are crucial for the oxygen and nutrient transport in the organism that enables a proper development and maintenance of the perivascular tissue homeostasis [65]. Furthermore, the vascular network plays an indispensable role in draining metabolic waste and, upon request, activate thrombotic and inflammatory responses [66]. During the adulthood, new branches are mostly generated by the tightly cooperated action of leading tip cells, that migrate into avascular region, and the stalk cells, that proliferate to follow the moving tip cells [67]. Once newly formed sprouts meet each other, they fuse to form a new vessel. Here, tip and stalk cells differentiate into the so called phalanx cells, and then the maturation of the vessel occurs with the recruitment of mainly pericytes an fibroblasts [68]. Within the tumor, excessive microvascular network is formed to supply the high request of a growing tumor mass characterized by a chronically active pro-angiogenic signature [69]. However, cancer-associated vessels often show structural and functional abnormalities. The tumor vasculature lacks the organized hierarchy of a normal vascular bed and instead comprises leaky, tortuous, and immature branches [70]. Therefore, the blood flow results chaotic and slow with a consequent inefficient cancer perfusion finally conveying the insurgence of persisting or intermittent hypoxic areas [69,71]. Accordingly, the TME becomes subjected to HIF that drives transcriptional responses promoting adaptation and selection of both cancer and stromal cells, thus inducing pro-survival changes [72]. In this scenario, EC metabolic alterations crucially affect TME composition and function. Here, we will focus our attention on relevant TGF-β mediated processes involving ECs.

### 3.2. How Does TGF-β Directly Alter EC Metabolic Reprogramming?

ECs demonstrate a high plasticity, with several tunable and dynamic phenotypes switching from Tip, Stalk and Phalanx cells. Beside different names, these multiple functions reflect different proliferative and biosynthetic requirements associated with precise metabolic state and demands [1]. In migrating tip cells, for instance, the rapid energy supply required for fast moving is guaranteed by high glycolytic flux which can produce more ATP in a shorter time with respect to oxidative metabolism [73]. Here, 85% of the total cellular ATP content relays to glycolysis [74]. Indeed, upon pro-angiogenic factor stimulation, GLUT1 and glycolysis regulator 6-phosphofructo-2-kinase/fructose-2,6-bisphosphatase 3 (PFKFB3), an activator of phosphofructokinase 1, are upregulated at lamellipodia and filopodia with the consequent increased in glycolytic flux [74]. Concomitantly, the pro-stalk signaling Dll4 starts to be highly expressed on tip cells. Dll4 activates Notch signaling on the near stalk cells prompting their differentiation and delineating their metabolism. Indeed, the Dll4-dependent cleavage of the Notch intracellular domain (NICD) transcriptionally downregulates the PFKFB3-driven glycolysis and conversely enhances fatty acid (FA) binding proteins (FABP) expression [74]. FABP-dependent FA metabolism is essential for EC proliferation, a key function of stalk cells. Notably, FAs are loaded into stalk cells by a “flip-flop” mechanism or through specific transport proteins, such as CD36 and FA transport proteins [75]. The rate-controlling step of stalk cell FA β-oxidation (FAO) depends on the activity of the carnitine palmitoyltransferases (CPTs) that shuttles the activated fatty-acyl-CoA synthase (FA-CoA) to mitochondria. This process replenishes the tricarboxylic acid (TCA) cycle and leads to the production of aspartate as a precursor for deoxyribonucleotide triphosphate (dNTPs) for cell replication [76]. Despite being deeply characterized for tip and stalk cells, phalanx cell metabolism is still poorly elucidated. However, a central role has been recently given to FOXO1 that belongs to the forkhead box O (FOXO) proteins, a sub-group of the FOX transcription factor family [77]. FOXO1 acts inhibiting Myc, a key transcriptional factor in growth and anabolic metabolism, to decrease glycolysis and impair mitochondrial function, thus ultimately inducing EC quiescence [78]. Acetyl-CoA produced by FAO enters the TCA cycle, with the consequent antioxidative NADPH for the maintenance of barrier integrity. In this line, oxidative pentose phosphate pathway (oxPPP) produces additional NADPH exploitable to regenerate antioxidant reduced glutathione (GSH) [66]. TGF-β and BMP pathways both have effects on angiogenesis through the ALK-1 interaction, which is specifically expressed in vascular EC. They can bind to ALK-1 activating the EC proliferation via pSmad1/5 signaling [79]. Conversely, different endothelial activation states reflect changing in endothelial TGF-β release. In particular, it has been reported that sprouting endothelial tips are characterized by reduced TSP-1 transcription and enhanced expression of pro-tumor factors POSTN and TGF-β1 [80]. By contrast, a stable microvasculature constitutes a dormant niche described by high level of TSP-1, a potent regulator of latent TGF-β activation [81]. Beside the conventional states, EC can be triggered toward an endothelial-to-mesenchymal transition (EndoMT) under TGF-β control during tumor progression [82]. This process involves the activation of a trans-differentiation program whereby ECs escape their endothelial characteristics and gain mesenchymal behavior. EndoMT promotes cancer development and metastasis, but also influence the response to cancer therapy [83]. Indeed, the EndoMT cytoskeletal reorganization through the Rho/ROCK signaling pathway linked to the loss of endothelial adhesion molecules (VE-cadherin, claudins) contributes to the disruption of the endothelial barrier favoring the intra- and extravasation of tumor cells [82]. Furthermore, the aberrant vessel maturation, the decreased expression of vascular endothelial growth factor receptor (VEGFR) and the hypoxia promotion concur to vanish anti-angiogenic strategies and confer resistance to anti-tumoral therapies [82].

During EndoMT, EC undergo morphological, functional and molecular changes, including a decline of their adhesion molecules, increase expression of mesenchymal biomarkers, and conversion of their apico-basal polarity in favor of a front-end back polarity. TGF-β induces EndoMT alterations by involving the Snail family of transcription factors such as Snail, Slug, Twist, and ZEB [84]. Snail family activation can occur through: (i) the direct action of TGF-β involving both Smad-dependent and PI3K/p38 MAPK-dependent signaling pathways [85,86] and (ii) through the hypoxia mediated phosphorylation of the transcription factor Twist-1 [87]. Moreover, activation of the EndoMT program has been recently associated with a metabolic alteration. Indeed, a decreased expression of CPT1 and a decline of acetyl-CoA levels conduits to both the inhibition of FAO, and the associated EndoMT [88]. Again, TGF-β participates to this metabolic-driven EndoMT response by impairing SMAD7 acetylation and signaling [88,89].

### 3.3. Does the EC Activity Shape the Tumor Metabolism?

As described, multiple events altering EC metabolism in tumors crucially impinge on cancer fate. It has to be considered that cancer cells and ECs reciprocally stimulate each other’s behavior. Indeed, whether angiogenesis occurs to promptly feed the metabolically demanding cancer cells, these latter enhance metabolism of ECs through releasing pro-angiogenic factors. The establishment of this positive feed-forward loop promotes both vessel growth and tumor development [90]. The constant secretion of pro-angiogenic/pro-glycolytic growth factors such as vascular endothelial growth factor (VEGF) by tumor cells in combination with a hypoxic microenvironment generated by both tumor and stromal cells, modify the EC expression of glycolytic enzymes. Indeed, ECs display higher rates of glycolysis than healthy ECs. Notably, pro-inflammatory cytokines further sustain this process. In response to cytokine activation, such as interleukin (IL)-1β stimulation, IκBα becomes phosphorylated before degradation, resulting in release and activation of p65 and p50. Thus, a cytokine-mediated signaling via NF-κB causes the release of pro-angiogenic factors by cancer, stromal and immune cells [91]. All these pathways participate to the upregulation of genes involved in pyruvate metabolism and glucose transport in hypoxic ECs. In addition to the direct effect of cytokines and hypoxia on ECs, tumor cell metabolism directly reprograms the metabolic pathways of ECs. Indeed, cancer-catalyzed lactate that accumulates in the stromal microenvironment might be internalized by ECs sustaining pyruvate oxidation [92]. Moreover, once uptaken, tumor-derived lactate induces a ROS-mediated NF-κB and IL-8 induction in hypoxic ECs [93]. Furthermore, lactate influx seems to participate in modulating receptor tyrosine kinase expression (among them also VEGFR2) [94] and in PHD2/VHL-dependent degradation [95] thus finally amplifying the VEGF signaling. Beside the direct and indirect effects of tumor cell on EC metabolism, tumor-associated macrophages (TAMs) can also influence EC glycolysis. Indeed, hypoxic TAMs strongly upregulate the expression of REDD1, a negative regulator of mTOR. Consequently, REDD1-mediated mTOR inhibition blocks glycolysis in TAMs avoiding the metabolic competition with ECs. The accumulation of glycolytic substrates, uniquely exploitable by highly demanding ECs sustains the excessive angiogenic responses, with consequent formation of abnormal blood vessels [96].

## 4. Immune Cells Tgfb-Induced Metabolism

### 4.1. How Does TGF-β Make-Up the Immune Landscape of the Tumor Microenvironment?

Immune cells represent a characterizing feature of the tumor landscape. Indeed, distinct cell populations, belonging to both innate and adaptive immunity, early and progressively settle the TME where they start a dynamic crosstalk with tumor cells and the surrounding environment. This tangled interplay sets off a peculiar milieu of chemokines, cytokines and metabolites that in turn, has a great impact on tumor development [97]. Seminal works have contributed to dissect the functional significance of cytokines and chemokines in cancer [98,99] and more recently, other studies focus on their relevance for cancer therapy [100,101]. Among the plethora of TME-related factors, TGF-β has drawn increasing attention over the last years. Although recognized as a tumor suppressor factor for healthy cells and early-stage cancer cells, in late-stage cancer TGF-β becomes the archetype of the tolerogenic tumor milieu, by primarily controlling the commitment and effector functions of immune cell subsets within the TME [102,103]. In this regard, TGF-β signaling represents a negative hallmark for many tumors and its activation has been linked with poor prognosis in colorectal [104] lung [105], breast [106] cancers, among others. Fundamentally, TGF-β can act by directly inhibiting anti-tumor immunity while sustaining the phenotype and function of pro-tumoral immune cells. As main effect, it shut down the differentiation and activity of Th1 T lymphocytes, that constitute the most effective immune responders against cancers. Clear evidence showed that the selective ablation of TGF-β signaling, in both CD4^+^ and CD8^+^ T cells, results in the activation of effective anti-tumor immunity in pre-clinical mouse tumor models [107]. Of note, in the promising era of anti-checkpoint inhibitors for the treatment of not curable tumors, TGF-β poses a major hurdle. Indeed, the presence of TGF-β in the TME associated with desert tumors, and resistance to checkpoint inhibitor treatment [108]. TGF-β blocked cytotoxic T lymphocytes (CTLs) by specifically inhibiting the expression of perforin, granzyme A, granzyme B, Fas ligand, and interferon-γ molecules that are collectively responsible for CTL-mediated tumor cytotoxicity [109]. Importantly, TGFβ-associated pathways emerged as relevant signaling in patients who were non responsive to anti–programmed cell death-1 (PD-1) therapy in both melanoma and metastatic urothelial cancer [110,111]. Specifically, the exclusion of CD8^+^ T cells from the tumor parenchyma correlated with signature of TGF-β signaling activation in fibroblasts in metastatic urothelial cancer patients, who were treated with an anti-PD-L1 agent (atezolizumab) [111]. In addition, pre-clinical studies highlighted a strong association between TGF-β activation and unresponsiveness to anti-PDL1 therapy, further suggesting that combined approaches targeting TGF-β pathway and checkpoint blockade might represent a promising strategy for the treatment of non- responsive tumors [111,112].

By triggering the expression of FOXP3, the master transcription factor of the TGF-β program, TGF-β signaling plays a cardinal role for the induction and activity of regulatory T cells (Tregs) that are highly enriched in tumors [113,114]. Of note, increased TGF-β level within the TME, which is common in late-stage cancers [115] leads to a frequent skewing or trans-differentiation of T cell phenotype toward the regulatory one [116,117]. In turn, the Treg-dependent upregulation of TGF-β by cancer cells expedite their invasiveness in a pre-clinical mouse model of melanoma [118]. The tight connection between TGF-β Signaling pathways and Tregs in tumors has been eminently dissected in other works [119].

Among the myeloid cell repertoire, macrophages represent one of the major components of immune cell infiltrate populating the TME. Importantly, the functional polarization of resident and recruited macrophages toward an immunosuppressive pro-tumoral phenotype, commonly defined TAMs, has been widely described in multiple human tumors. Conventionally, TAMs are strong predictor of metastasis and poor prognosis for patients in multiple cancer type [120]. Although initially defined as a macrophage-deactivating agent [121], the role of TGF-β in regulating macrophage fate has been progressively updated. As a matter of fact, TGF-β tunes macrophage phenotype and functions in vitro but it also plays a determinant role in regulating macrophage phenotype in vivo [122]. The involvement of TGF signaling in the development of TAMs has been extensively investigated in the last decade [123,124]. Multiple line of evidence indicated that this cytokine instructs macrophage to activate pro-tumoral programs and sustain the metastatic spreading. In lung cancer, the TGF-β mediated induction of IRAK-M in TAMs has been described as a key mechanism promoting tumor growth and immune tolerance [125]. It was also shown that TAMs sustained gastric cancer cell invasion and metastasis through the TGFβ2/NF-κB/Kindlin-2 axis [126,127].

Interestingly, TAM activity supports tumor metastasis being directly involved in the promotion of EMT and the formation of the pre-metastatic niche. In this regard, the released of several factors by TAMs, and among them TGF-β, is crucial for cancer cell migration and invasion [128].

Mounting evidence has shed light on the immunomodulatory role of an additional myeloid cell subset within the TME defined as tumor-associated neutrophils (TANs) [129]. TANs can exert either antitumor responses by the direct killing of tumor clones or sustain tumor development and spreading by promoting immunosuppression, angiogenesis, and ECM remodeling [129]. Remarkably, although less investigated, TGF-β signaling has been linked to neutrophil functions as it can act as chemoattractant for this cell subset in distinct chronic disease settings [130]. In vitro data showed that TGF-β can inhibit neutrophil activity and cytotoxicity [131]. Following this observation, the same group found that in tumors TGF-β sustains the generation of TANs with a protumoral phenotype (N2); on the other hand, the pharmacological blockade of TGF-β promotes the recruitment and activation of antitumoral (N1) TANs in two different tumor types (NSCLC and mesothelioma) finally leading to enhanced anti-tumor immunity in tumor-bearing mice [131]. Paralleling data were proposed in colon rectal cancer; the use of anti-TGF-β (1D11) in vitro promoted the functional skewing of neutrophils, that upon coculturing with tumor cells, increased their cytotoxicity and decreased the release of pro-metastatic factors [132]. Through the same mechanisms, the treatment with anti-TGF-β reduced tumor growth in vivo in an azoxymethane (AOM) and dextran sulfate sodium (DSS) induced colon rectal cancer mouse model [132]. More recently, a novel mechanism of T cell suppression caused by TGF-β-mediated neutrophil activation has been described in mouse model of colon adenomas [133]. The same study also investigated the clinical significance of the concomitant TGF-β activation and neutrophil recruitment for human colon-rectal cancers, thus suggesting novel therapeutic targets for the disease [133].

The role of myeloid suppressor cells (MDSC) in cancer progression has been largely documented [134]. This population of atypical bone-marrow cells including myeloid cells at early stages of differentiation, monocyte/macrophage precursors, granulocytes, gradually expand in tumor-bearing hosts; MDSC circulate in the blood and populate primary tumor tissue where they activate pro-tumoral programs mainly aimed at circumvent anti-tumor immunity. Being TGF-β an abundant player of tumor tolerance, its connection with MDSC biology has been recently investigated. TGF-β signaling pathways are linked to MDSC biology in multiple ways [135]. Although a general consensus appointed TGF-β as an inducer of MDSC immunosuppressive phenotype and activity, controversial reports can also be found. In addition to a direct effect on MDSC maturation from immature bone marrow precursors, TGF-β can indirectly tune MDSC activity in tumor bearing mice by inducing microRNA (miR494) expression. Importantly, miR494 ablation in MDSCs slow down tumor growth and metastasis [136]. An interesting report showed that TGF-β acts as a potent enhancer of monocytic MDSC (Mo-MDSC) expansion and immunosuppressive functions when combined with distinct MDSC-related cytokine stimulation [137]. Although the best characterized activity of MDSC is the suppression of T cell responses, they can directly act on innate immune components. In this regard, it has been shown that MDSC impaired natural killer (NK) cell responses in liver cancer-bearing mice by inhibiting NKG2D expression, NK cell-mediated cytotoxicity, and IFN-γ production through their membrane-bound TGF-β1 [138]. On the opposite site, a different study showed that MDSC generated from ex vivo isolated human peripheral blood mononuclear cells (PBMC) reduced their immunosuppressive potential on T cell proliferation upon TGF-β1 stimulation [139]. More recently, an immune-stimulating and tumor killing ability has been attributed to MDSC generated by bone marrow progenitors in the presence of TGF-β1 (TGFβ-MDSC). Although totally unexpected, authors showed that the adoptive transfer of TGFβ-MDSC in combination with radiotherapy lead to in vivo tumor regression and long-term survival of mice subcutaneously. Injected with MEER murine pharyngeal epithelial cells [140].

### 4.2. How Does Immune-Metabolism Shape the Tumor Microenvironment?

Seminal works unequivocally identified the metabolic reprogramming of tumor cells as a ground hallmark of cancer progression [70]. Later on, metabolic adaptations have been also described as crucial determinant for immune cell fate, controlling the phenotypic and functional behavior of both lymphoid and myeloid cells [141,142,143]. As immune cells populated the TME, it was reasonable to question whether metabolic reprogramming in tumors might affect immunity driving either pro- or anti-tumor responses. Pioneering the idea that metabolism might determine immune functions, several reports showed that T cells efficiently commute their metabolic programs depending on their differentiation and activation state. This concept has been authoritatively descanted over the last years [143,144].

As we reviewed, the metabolic switch occurring in transforming tissues has a great impact on tumor-infiltrating T cell functions [145]. In the following paragraphs we will focus on key metabolic events that specifically involve T cells within the TME.

The poorly-vascularized environment, the high glycolytic rate of proliferating tumor clones frequently caused nutrients/amino-acids depletion and pH perturbation within the TME. Interestingly, the number of cytotoxic effector cells was increased in melanoma-bearing mice in which lactate production was significantly lowered by the silencing of the lactate dehydrogenase A enzyme [146]. Indeed, a pathological level of lactic acid resulted in reduced IFN-γ production affecting nuclear factor of activated T cells (NFAT) signaling pathway. Of note, a negative correlation between LDHA expression and T cell activation markers was described in in human melanoma patients [146]. Glucose availability also impinges on NFAT activation in both CD4^+^ and CD8^+^ T cells by affecting the production of the glycolytic metabolite phosphoenolpyruvate (PEP) through the regulation of the calcium sarco-endoplasmic reticulum calcium ATPase (SERCA) channel [147].

A general unbalance of the amino acid equipment is a general feature of multiple tumor environments. Among them, arginine, glutamine and tryptophan deprivation within the TME has been linked to T cell suppression and tolerogenic programs. In particular, the increased activity of the Indeolamine-2,3-dioxygenase (IDO), converting tryptophan to kyneurinine, has been described in different human cancer, including melanoma and ovarian cancer [148]. Sustained IDO activity led to a reduced tryptophan availability that in turn impaired effector T cells activation and proliferation by TORC1 inhibition [149]. The accumulation of kyneurinine contributes to ineffective TCR signaling and consequent T cell dysfunction, impaired proliferation, and cell death. In contrast, kyneurinine is a ligand of the aryl hydrocarbon receptor (AhR) and may enhance Treg differentiation [149,150]. A striking recent report shed new light on the previous knowledge on glutamine metabolism in tumors and its impact on antitumor immunity. Authors showed that glutamine antagonism enhanced the activation and survival of effector T cells and boosted the efficacy of PD-1 checkpoint immunotherapy in multiple tumor mouse models [151] thus establishing glutamine metabolism as a metabolic checkpoint for cancer immunotherapy. By interfering with glutamine metabolism, authors suggested the possibility of differentially modulating the metabolism of cancer cells and antitumor immune cells by exploiting the peculiar metabolic plasticity of the distinct cell type [151].

### 4.3. How Does TGF-β Influence Immune-Metabolism in the Tumor Microenvironment?

As previously mentioned, multiple metabolic checkpoints control anti-tumor immunity. In this regard, TGF beta has been kept under special watch as it directly fuels tolerogenic programs by the metabolic regulation of TME players. The investigation of TGFβ-dependent metabolic reprogramming of immune cells within the TME represents an emerging field of tumor biology with important clinical implications. Recently, it has been proposed a duration-dependent effect of TGF-β stimulation on CD4^+^ T cells [152]. In particular, authors showed that TGF-β significantly inhibited the basal and adenosine triphosphate (ATP)–coupled oxygen consumption rate (OCR) and decreased the function of complex V, ATP synthase, leading to impaired IFN-γ production by CD4^+^ cells [115]. Interestingly, a prolonged TGF-β CD4^+^ T cell stimulation (72 h), to push the Treg commitment, increased mitochondrial membrane potential and spare respiratory capacity [152]. Upon TGF-β stimulation, Treg showed a high oxidative metabolism, limiting glucose metabolism by lowering the expression glycolytic genes, such as GLUT1 and Hexokinase 2 and promoting the inhibition of the TCR-CD28-PI3K-mTOR pathway). At the same time, they can enhance FA oxidation to provide intermediates for TCA cycle [153].

As for T lymphocytes, a similar time-dependent effect has been also described for TGF-β stimulation of NK cells. Indeed, 18 h- stimulation with TGF-β significantly decreased the rate of IL-2–induced mitochondrial metabolism in NK cells that downregulated the expression of CD69, CD71, IFNγ and granzyme B without affecting mTORC1 activity [154]. In contrast, prolonged TGF-β stimulation (5 days) inhibited the mTOR-dependent metabolic activity NK cells [155].

The immunosuppressive role of TGF-β1 with respect to NK cell activity has been largely documented. As main effect, TGF-β1 promoted the down-regulation of NKG2D expression in both autocrine and paracrine manner, suggesting that it can act as negative regulator of NK cell effector function [156,157] but it can also prevent NK cell overactivation [158]. Mechanistically, TGF-β1 enhanced the expression of mature miR-1245, leading to down-regulation of the NKG2D receptor in NK cells [159].

As mentioned above, TAMs represent one of the leading players fostering the malignant potential and invasiveness within the tumor ecosystem. TGF-β has been very recently appointed as a relevant metabolic regulator of the interplay between TAMs and breast cancer cells [160]. Indeed, the high level of lactate produced by breast tumor clones in the TME increased the secretion of the chemokine CCL5 by the activation of Notch signaling in human macrophages. Further, lactate-activated macrophages produced large amount of TGF-β that in turn, promote breast cancer cell migration and EMT via CCL5-CCR5 axis [160]. More recently, it has been showed that TAM-derived TGF-β1 enhanced breast tumorigenesis by controlling the expression of the succinate dehydrogenase enzyme in breast cancer cell through the transcriptional regulation of STAT1 activity [161]. Conventionally, anti-inflammatory macrophages rely on mitochondrial OXPHOS to exert their functions [162]. Park and colleagues recently showed that exosomes- derived from different tumor cell lines cultured in hypoxic conditions- were highly enriched in multiple immunomodulatory factors including TGF-β and let-7a miRNA; these exosomes were effective in promoting infiltrating myeloid cell polarization toward the M2-macrophage phenotype and boosting their activation effector functions by enhancing OXPHOS through the reduction of AKT and mTOR phosphorylation [163]. The hypoxic tumor microenvironment also resulted in increased macrophage recruitment and M2-type phenotype skewing in gliomas. Specifically, TGF-β enhanced the expression of the tumor promoting factor periostin via the RTK/PI3K pathway in U87 and U251 glioma cells cultured under hypoxic conditions [164].

As a major immunosuppressive component of the TME, MDSC has drawn increasing attention over the years. In particular, an emerging field is represented by the impact of MDSC metabolism on their commitment and tolerogenic potential in tumor-bearing hosts. The activity of MDSC is controlled by multiple metabolic parameters such as oxygen/nutrient availability, pH, metabolite levels, and reactive species within the TME. MDSC can metabolically sense and adapt to nutrient cues of the surrounding environment, activating the most efficient metabolic pathway according to their needs. MDSC metabolism mainly stands on glycolysis but they can also exploit OXPHOS, salvaging TCA cycle, FAO and lipid metabolism as energy metabolic pathways to sustain their development and activity in tumors [165,166]. On the other hand, MDSC and TAMs undertake distinct metabolic pathways to suppress anti-tumor immunity and foster tumor progression. In this regard, cumulative evidence has identified leading pathways controlling T cell responses by MDSC/TAM metabolic interference [167]. We and other have previously shown that MDSC, through the depletion of amino acids, as arginine and tryptophan, impaired the recruitment and the immune functions of anti-tumor T lymphocytes in the TME [168,169]. In addition to other cytokine secreted by MDSC as IL-10, TGF-β sustained tumor growth by specifically interfering with antitumor immunity and fostering tolerogenic programs [170]. MDSC undergoing metabolic reprogramming up-regulated the expression of two enzymes, CD39 and CD73 that catabolize ATP molecules generating extracellular adenosine, a well-recognized modulator of anti-tumor responses [171,172]. In this regard, it has been reported that the sustained activation of ATP metabolizing enzymes in MDSC from non-small cell lung cancer (NSCLC) patients is triggered by the TGF-β-mTOR-HIF-1 signaling [173]. Specifically, TGF-β stimulation promoted the phosphorylation of mTOR by Smad2/3-dependent pathway. More, rapamycin treatment, to specifically interfere with the mTOR pathway, abrogated the TGF-β-mediated induction of CD39/CD73 expression suggesting a putative role for autophagy in the regulation of the two enzymes in MDSC in NSCLC [173]. The increase of aerobic glycolysis by tumor cells alters the expression level of GM-CSF and G-CSF, which are essential cytokine for MDSC development [174]. In this fashion, evidence indicated that TGF-β1 contributed to the control of the transcriptional activity of C/EBPβ, a master regulator of MDSC sustenance. In particular, the repression of the AMPK-ULK1 signaling by the high glycolytic rate of triple negative breast cancer cells, reduced the autophagy-mediated degradation of C/EBPβ-LAP isoform; in turn, LAP enhanced G-CSF expression further supporting MDSC expansion in tumors [174].

Our group has been actively involved in the characterization of MDSC immunosuppressive properties in particular focusing on the dysregulation of arginine metabolism in tumors [175]. As well documented, amino acid shortage features the TME landscape. In particular, arginine and tryptophan depletion has been linked to the generation and expansion of immunomodulatory myeloid cell subsets within the TME. In macrophages, TGF-β has been shown to upregulate the Arginase (ARG) enzyme [176] which represents, together with nitric oxide synthases (NOS), a major arginine catabolizing enzyme. In cancer, the expression and activity of both enzymes has linked to the immunosuppressive function of MDSC and M2/TAMs [177]. In this regard, the partial uncoupling of the NOS activity caused by l-arginine shortage within the TME lead to the generation of reactive nitrogen species (RNS) that further amplificated MDSC-mediated tumor immunosuppression. The nitration of relevant proteins including T cell receptor [178] and the CCL2 chemokine [169] specifically blunted T cell responses by either inhibiting tumor antigen recognition or intratumoral CD3^+^ cell recruitment.

Among the tolerogenic cell population in the TME, TANs are attracting progressive attention. Conventionally, neutrophil metabolism relies on glycolysis due to the limited number of mitochondria in these cells [179]. However, under glucose deprivation, neutrophils are able to reprogram their metabolism toward FAO and glutamine metabolism to obtain energy and production of intermediates for Krebs cycle [180]. Importantly, the pentose-phosphate pathway (PPP) is often overrepresented in neutrophils as a primary way to obtain NADPH, a substrate for NOX2 enzyme that allows ROS production during neutrophil activation [181]. TANs, that under TGF-β exposure undertake N2 polarization [131] are metabolic characterized by excessive ROS production and high level of Arginase 1 enzyme expression; in turn, ARG1 activity contributes to blocking T cell activation in tumors by lowering arginine availability in the extracellular environment [182]. Remarkably, SLC27A2, solute carrier family 27 (FA transporter) member 2, regulated by STAT5 activation, also contributed to TAN metabolic reprogramming in tumors [183]. More studies will be required to specifically investigate the contribution of TGF-β signaling to TAN metabolism in tumors.

## 5. Conclusions

TGF-β represents a pleiotropic cytokine that plays crucial and non-redundant roles in the control of immune system functionality in health and diseases. Cumulative evidence over the years has clearly showed that the activation of TGF-β can flow to either pro- or anti-inflammatory responses depending on the peculiar environment in which it is acting. The duality of TGF-β signaling is well-recognized in cancer where the triggering of the canonical Smad pathway, but also the non-canonical MAPK-GTPase one, might lead to both tumor promotion and suppression. Despite a direct effect on tumor clones themselves, TGF-β, significantly contributed to tumor development by acting on stromal neighbors populating the TME; it pushes the commitment and functions of multiple cell type, fostering at the foremost the development of immunosuppressive populations and the establishment of tolerogenic circuits in tumors (Figure 2).

Over the last decades, cancer metabolism has been appointed as an emerging determinant for tumor growth and spreading. Among the plethora of factors contributing to shape the metabolic landscape of the TME, TGFβ, which is largely represented in several tumors, has aroused progressive interest. As matter of fact, TGF-β sustained the metabolic reprogramming of CAFs, EC and innate and adaptive immune cells populating the primary tumor bulk by activating multiple tolerogenic pathways. Thus, the identification of metabolic checkpoints under TGF-β control within the TME would represent a productive source of novel targets for therapy. This would be extremely worthwhile for cancer immunotherapy that needs to face and overcome immunosuppressive barriers in the host to be fully effective. Combinatory approaches, targeting immune checkpoints and the TGF-β cascade, have been undertaken showing encouraging pre-clinical efficacy. Although the field is progressively growing, many outstanding questions still remain to be investigated. In particular, the need to manage the pleiotropic effects of TGF-β in the hosts that are prominently dependent of timing and cytokine dosage. In this regard, the identification of key metabolic nodes under TGF-β control might enhance therapeutic outcomes by targeting selective pathways nourishing the tolerogenic environment that dismantle anti-tumor immunity in the hosts.

## Figures and Tables

**Figure 1 cancers-13-00401-f001:**
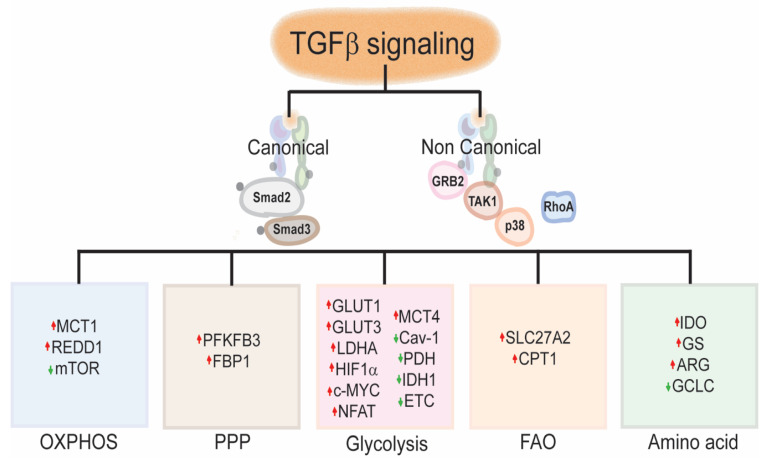
TGF-β signaling and metabolic reprograming: TGF-β canonical and non-canonical signaling activation promotes the metabolic skewing to either glycolytic or oxidative metabolism in the TME by modulating the transcription of genes involved in glycolysis, OXPHOS, PPP, FAO or amino acid metabolism, directly or indirectly (see text.) TGFβ, Transforming Growth factor beta; GRB2, Growth factor receptor-bound protein 2; TAK1, Mitogen-activated protein kinase kinase kinase 7; p38, p38 mitogen-activated protein kinases; RhoA, Ras homolog family member A; MCT1, Monocarboxylate transporter 1; MCT4, Monocarboxylate transporter 4; REDD1, Regulated in development and DNA damage response 1; mTOR, mechanistic target of rapamycin; PFKFB3, 6-phosphofructo-2-kinase/fructose-2,6-biphosphatase 3; FBP1, Fructose-1,6-biphosphatase, GLUT1, Glucose transporter 1; GLUT3, Glucose transporter 3; LDHA, Lactate dehydrogenase A; HIF1α, Hypoxia-inducible factor 1 alpha; c-MYC, MYC proto-oncogene BHLH Transcription factor; NFAT; Nuclear factor of activated T-cells; Cav-1, Caveolin 1; PDH Pyruvate dehydrogenase; IDH1, Isocitrate dehydrogenase 1; ETC, Electron transport chain; SLC27A2, solute carrier family 27 (fatty acid transporter), member 2; IDO, Indoleamine 2,3-dioxygenase; GS; Glutamine synthetase; ARG, Arginase; GCLC, Glutamate-cysteine Ligase catalytic subunit; OXPHOS, Oxidative phosphorylation; PPP, Pentose Phosphate Pathway; FAO, Fatty acid oxidation.

**Figure 2 cancers-13-00401-f002:**
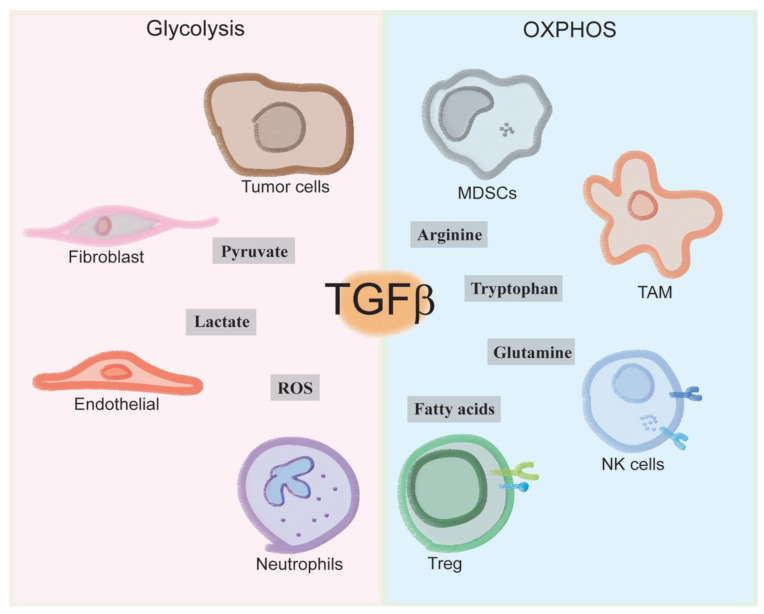
Overview of metabolic reprograming by TGF-β: TGF-β signaling sustains immune evasion by orchestrating the metabolic reprogramming within the TME. Tumor cells, fibroblasts, EC and neutrophils display high rates of the glycolytic metabolism, releasing lactate, pyruvate and producing high amount of Reactive Oxygen Species (ROS). MDSCs, macrophages (TAM), NK cells and Treg cells preferentially activate the OXPHOS to get energy, enhance FA oxidation and amino acid metabolism as a source of intermediates for the Krebs cycle, fostering the generation of a tolerogenic environment that contributes to inefficient anti-tumoral immune activation.

## Data Availability

No new data were created or analyzed in this study. Data sharing is not applicable to this article.

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
