# Peer review of "TGF-β in Cancer: Metabolic Driver of the Tolerogenic Crosstalk in the Tumor Microenvironment"

_cancers, 2021, doi:10.3390/cancers13030401_

Round 1
Reviewer 1 Report
In this comprehensive review article the authors have summarized recent literature on the role of TGF beta in regulating tumor microenvironment, with an emphasis on the stromal and immune cell population. Overall, the manuscript is well organized and well written. There are a few comments from this Reviewer:
In each type of cells regulated by TGF beta, it is unclear regarding the source of this cytokine: autocrine, paracrine, etc.
Mild English editing is suggested to improve the quality of this manuscript. For instance:
Line 12: “a driver” instead of “driver”
Line 22: “a key” instead of “key”
Author Response
We really thank the reviewer for the constructive comments and suggestions. We revised the manuscript accordingly (changes are marked in red throughout the text). We hope that the new version of the manuscript will met all your requirements.
Reviewer 2 Report
The review manuscript TGF-β in cancer: metabolic driver of the tolerogenic crosstalk in the tumor microenvironment is a valuable work that adequately covers roles of TGF-beta and its differential effects on cell and tissue metabolism in tumor microenvironments. The sectional organization of the review will inform even uninitiated readers of the ideas underlying metabolic reprogramming that is driven by TGF-B in the tumor microenvironment.
Processes of glycolysis vs. oxidative phosphorylation in transformed cells and tissues is explained succinctly, as is the roles of TAMs and TANs in metabolic remodeling of the tumor microenvironment.
Listed below are several suggestions that if completed will improve the effectiveness of the review.
Parts of the review will require careful editing before it is at a publication level.
Management of abbreviations is heterogenous. Most of the abbreviations are spelled out and a functional description is provided, which is an informative, reader-friendly provision that the review provides. However, there are numerous instances where the spelling and function are omitted, and others where the spelled-out name is redundant (ie, see EMT, lines 85, 143, and 364. Each abbreviation should be checked.
Line 12, L52 and elsewhere: The spelled-out name for TGF-B should be “Transforming” as opposed to “Tumor.”
L63: Integrins should be referred to as cell-surface transmembrane receptor proteins as opposed to ECM proteins.
L64: Most, if not all of the alpha-v integrins have been shown in vitro to bind and activate latent TGF-beta via the RGD in the latency-associated peptide.
L79: The sentence is repeated, almost literally, in L93.
L342: A citation should be included for the increase of TGF-beta in the TME of certain late-stage cancers.
L380: Explaining the AOM/DSS model of CRC, and the advantage of driving TANs to an anti-tumor phenotype would be useful.
L399: Define Mo-MDSC
L550: SLC27A2 (a lipid acid transporter) differs from the description of line 113: SLC27A2 (Solute carrier Family 27 member 2).
Author Response

(The authors gave the same response as above.)
